# Impacts of House Mice on Sustainable Fodder Storage in Australia

## Peter R. Brown *  and Steve Henry

CSIRO Health & Biosecurity, GPO Box 1700, Canberra, ACT 2601, Australia; Steve.Henry@csiro.au

* Correspondence: Peter.Brown@csiro.au

**Abstract:** Mice cause substantial economic, social, and environmental damage to rural production systems and communities in Australia, especially during mouse plagues. The losses and damage caused by mice to hay/fodder storage are not well understood, given the size of the industry, so a pilot study (survey of 14 growers) was designed to better understand the physical and economic damage, consider disease implications, and identify the management strategies available. During a mouse plague, mice were regarded as the only factor (other than weather) that affected the long-term sustainability of fodder storage. Growers were feeding fodder to livestock (sheep/cattle) at twice the rate they normally would because of mouse damage and contamination. Mice damaged strings and the structure of bales, making transport impractical, leading to destruction of some stacks. Losses caused by mice were estimated at AUD 140,000 (range = AUD 7000–461,580; equivalent to 30–40% loss of value) and included estimates of physical damage to hay/fodder bales, rodenticides, and labour costs. Growers were concerned about contamination and disease transfer to livestock (and workers) from mice through urine/faeces and their carcasses, especially during mouse plagues. There are significant gaps in our knowledge on the impacts of mice to fodder storage in Australia. Research is needed to (1) identify effective mouse control options, (2) determine the economic impact of mice, and (3) undertake a disease study of mice, to help inform appropriate management strategies for effective control.

**Keywords:** bales; damage; haystacks; livestock diseases; losses; management practices; mouse plague; *Mus musculus*; zoonotic diseases

## 1. Introduction

The house mouse (*Mus musculus*) is a serious pest to agriculture in Australia, with most damage occurring to pre- and post-harvest grain crops [1–6]. The house mouse has a worldwide distribution, and was thought to arrive in Australia with European explorers and settlers [7]. Mouse populations occasionally undergo widespread eruptions (mouse plagues or outbreaks) in the grain-growing regions of Australia [5], and develop in response to factors including mild weather and good rainfall, leading to increased food supply, reduced predation and diseases, and changes in social structure [8]. The success of the house mouse as a pest species can be attributed to its ability to live in a wide variety of habitats, its small size, neophilic behaviour (likes new objects and food types), its high reproductive potential, and omnivorous feeding habits [9].

The periodic outbreaks cause serious economic damage to grain crops and a range of other rural industries. In 1993/94, a mouse plague caused losses estimated at up to AUD 100 million [2], although, even in non-plague years, impacts from mice can be significant [2]. The 2011 mouse plague reportedly caused over AUD 200 million in crop damage alone and affected 3 million hectares of crops [10]. The mouse plague that affected central and northern New South Wales in 2021 has been estimated to cost AUD 1 billion, according to NSW Farmers (a farmer advocacy association) [11]. The NSW state Government set aside AUD 150 million in support for farmers and rural households primarily through rebates

for the cost of zinc phosphide rodenticide baits [12]. A significant area of concern through the mouse plague in NSW was the impact on hay and fodder storage.

The losses and damage caused by mice to hay and fodder storage are not well understood, especially during mouse plagues. Impacts to hay were identified during the 1993 mouse plague [2], including fouling that rendered the hay useless; however, losses were not quantified. It is noteworthy that there is no mention of rodent impacts to fodder in the second edition of Buckle and Smith's [13] "Rodent Pests and their Control" (considered a definitive collection of chapters on rodent impacts and management). This demonstrates the overall lack of existing knowledge of rodents in these systems and highlights the importance of collecting new information to help inform decisions by growers and support the industry. Impacts from mice go beyond direct physical damage to fodder, with the risk of disease transfer to livestock and farmers being a major concern [9,14].

In Australia, fodder is grown and cut to feed to livestock as green feed (hay) or harvested and dried (fodder) (the term "fodder" is used universally throughout this paper to represent hay or fodder). It is fed to livestock (dairy and beef cattle, sheep, pigs, horses, chickens, and rabbits) largely for weight maintenance, and/or to make up seasonal gaps between feed demand and supply during lean periods (e.g., drought). Fodder can be classified as legume hay (lucerne, vetch), grass hay (pasture grasses), cereal/grain hay (oaten, wheaten), mixed (forage hay oats, wheat, barley), and straw (for bedding in stables and also to provide bulk when mixed with higher value feeds such as lucerne hay, molasses, or grain to help it go further). It can be left in the field where it was grown, stacked along the edge of fields (with or without large plastic covers), or moved to large sheds. The fodder industry is a significant component of Australia's livestock industry, contributing some AUD 17.6 billion to Australia's GDP [15]. The fodder industry was estimated to be worth AUD 0.8–2.0 billion, and is larger than barley, sugar, and the poultry industries [16]. One-third of commercial farms in Australia produce fodder. Most fodder growers are grain growers or livestock producers, but also grow fodder as a component of their enterprise. Fodder can be stored for several years and is an integral part of risk management to offset variable seasons to ensure sufficient food resources are available. Most fodder is stored and used on the farm where it was produced, with some sold on local markets (e.g., feedlots) or traded internationally.

Given the infrequent nature of mouse plagues, the importance of fodder as a component of crop and livestock farming in Australia, and the dearth of knowledge about potential impacts of mice in fodder systems in Australia, it is important to better understand the range of impacts that mice have on sustainable fodder systems. This knowledge is required in order to consider what options are available to manage a potential mouse problem to minimise damage and losses and minimise disease risks for humans and livestock. Therefore, the aim of this paper is to review what is known about mouse impacts to sustainable fodder storage in Australia. This is based on a review of literature and from findings from a small survey of fodder producers, to develop recommendations for management and outline areas where research needs to be conducted.

## 2. Potential Impacts of Rodents on Fodder Storage

### 2.1. Disease Risk

Disease transfer to livestock and humans (including farm workers) is the main perceived risk of mice (and rats: *Rattus rattus* and *R. norvegicus,* which are also present in fodder but at low numbers). Disease transfer to livestock can occur through urine and faeces [17], but also through their carcasses. Rodents dribble their urine as they travel to help scent mark their territories, meaning urine is deposited everywhere, particularly along well-travelled routes. Diseases include murine typhus, Lassa fever, leptospirosis and angiostrongyliasis [14] (Table 1), and Q-fever [18]. Mice also carry a range of endo-parasites (nematodes, tapeworms, helminths), ectoparasites (ticks, mites, fleas), bacteria, and viruses (Table 1). Some 28 species of helminths have been recorded in house mice, with seven reported to infect humans [9]. This is not likely to be a definitive list, and more research

is needed, particularly now that screening techniques for identification of diseases has advanced rapidly in recent years.

**Table 1.** Summary of parasites, bacteria, and viruses recorded in wild populations of house mice (*Mus musculus*) in Australia. The parasites in bold are potential human pathogens. Source: modified from Singleton and Krebs [9] and references within.

| Taxon/Category | Species |
|---|---|
| Nematodes (Roundworms) | *Aspiculuris tetraptera* |
| | ***Calodium hepaticum*** |
| | *Gallegostrongylus australis* |
| | *Heligmosomoides polygyrus* |
| | *Heterakis spumosa* |
| | *Muspicea borreli* |
| | *Protospirura muris* |
| | *Syphacia obvelata* |
| | *Trichosomoides crassicauda* |
| | *Trichuris muris* |
| Cestodes (Tapeworms) | *Taenia taeniaeformis* |
| | *Rodentolepis diminuta* |
| | ***Rodentolepis fraternal*** |
| | ***Rodentolepis microstoma*** |
| Trematodes (Tapeworms) | ***Brachylaima cribbi*** |
| Acarina (Ticks and mites) | *Cheleytus* sp. |
| | *Echinonyssus butantanensis* |
| | *Eulaelaps* sp. *(stabularis ?)* |
| | *Kleemania lumosa* |
| | *Mesolaelaps australiensis* |
| | *Myobia murismusculi* |
| | *Myocoptes musculinus* |
| | *Ornithonyssus bacoti* |
| | *Paraspeleognathopsis bakeri* |
| | *Radfordia affinis* |
| | *Trichoecius rombousti* |
| Siphonaptera (Fleas) | *Nosopsyllus fasciatus* |
| | *Nosopsyllus londiniensis* |
| Bacteria | *Escherichia coli (E. coli)* |
| | *Salmonella* sp. *(Salmonella)* |
| | *Streptobacillus moniliformis* |
| | *Leptospira* sp. *(Leptospirosis)* |
| | *Clostridium botulinum (Botulism)* |
| Viruses | *Epizootic diarrhoea of infant mice virus (rotavirus) (EDIM)* |
| | ***Lymphocytic choriomeningitis virus (LCMV)*** |
| | *Mouse adenovirus (MAdV)* |
| | *Minute virus of mice (MVM)* |
| | *Mouse hepatitis virus (MHV)* |
| | *Mouse parvovirus (MPV)* |
| | *Murine cytomegalovirus (MCMV)* |
| | *Pneumonia virus of mice (PVM)* |
| | *Reovirus serotype 3 (Reo 3)* |
| | *Theiler's mouse encephalomyelitis virus (TMEV)* |

There are other diseases and parasites that can interact not just with livestock but also with domestic and feral cats (*Felis catus*) and dogs (*Canis familiaris*) (and foxes, *Vulpes vulpes*), which are also present on farms and likely to interact with mice. The potential risk associated with disease and parasites increases significantly during mouse plagues. Growers recognise disease as a problem, but the implications for human and livestock health are

not well understood. The three main concerns in Australia are botulism, leptospirosis and lymphocytic choriomeningitis virus (LCMV).

- Botulism is a disease caused by the botulinum toxin, which is produced by the bacterium *Clostridium botulinum* [19]. *C. botulinum* spores are common in the soil, and also in the gut of healthy normal cattle and other animals. Spores are the dormant form of the organism. Only the actively growing or "vegetative" *C. botulinum* bacteria produce botulinum toxin, and it is the toxin that produces the disease. *C. botulinum* spores will only germinate and grow under anaerobic conditions. Botulism outbreaks can occur in intensively fed beef and dairy cattle when the feed is contaminated by the botulism bacteria growing in rotting animals (e.g., dead mice) or vegetable material in the stored feed.
- Rodents are carriers of spirochetes of the genus *Leptospira* throughout the world and are important reservoirs of infection for humans and domestic animals [14]. It can lead to foetal abortion and stillbirths in livestock. Two of these strains (*L. hardjo* and *L. pomona*) are known to cause abortion in cattle, and there is a vaccine available to prevent this [20]. Humans acquire infection through consumption of food or water that is contaminated by rodents or by contact through skin or mucous membranes with soil or water (or contaminated hay or fodder) that is contaminated by rodent urine. Handling of dead infected rodents may also form a source of infection [14].
- Another significant disease is lymphocytic choriomeningitis virus (LCMV). It is a rodent-borne virus that can cause lymphocytic choriomeningitis (LCM) in humans, and causes flu-like symptoms through to meningitis and encephalitis [14]. It can also cause intrauterine infections leading to foetal death. Infections have occurred in Australia, and there was a report of a farmer that contracted the disease during the 2021 mouse plague in NSW (ABC news: https://www.abc.net.au/news/2021-04-04/mouse-plague-farmer-contracts-lymphocytic-choriomeningitis/100037256, accessed 16 November 2021). LCM is not a notifiable disease in Australia, so the full extent is unknown. It is likely the disease is spread through breathing air (dust from contaminated hay) that is contaminated with mouse excrements.

Extensive research into diseases of mice was conducted in Australia from the 1980s through until the early 2000s primarily in the search of a biocontrol agent and to understand the role of diseases in regulating mouse populations [21–23]. The main zoonotic diseases detected in Australia were *Streptobacillus moniliformis* (bacteria causing rat bite fever), *Angiostrongylus cantonensis* (roundworm causing rat lungworm), lymphocytic choriomeningitis virus (LCMV), leptospirosis (bacteria), rickettsia (bacteria transmitted to humans via the bites of fleas, lice, ticks, or mites) and cryptosporidia (parasite that causes diarrheal disease cryptosporidiosis). Singleton [24] found that the prevalence and mean intensity of endoparasite infections varied for each parasite species during a year when mice were consistently low in abundance and appeared to be independent of mouse population density. Fluctuations in the number of parasites appeared to be related to changes in rainfall and food availability.

While there is some information on what pathogens are present in Australian wild mice at the population level, there is little understanding on how pathogen composition varies with changes in mouse population abundance. While serological studies have shown that wild mice are regularly exposed to different viruses and parasites [22,25], these studies were often conducted at a single point in time and/or were targeted to predefined pathogens [21]. While wild animals frequently carry endemic microbes with no overt clinical disease, these may have significant population impacts by affecting fecundity [26], vulnerability to predation [27], or exacerbation of other stressors, such as poor nutrition or overcrowding [28]. Knowing which microbes are present in wild mouse populations and how these microbial communities change with different mouse population densities may provide insights into the role that pathogens play in regulating the boom-and-bust population cycles.

There have been significant advancements in next-generation sequencing technologies and bioinformatics pipelines. Metatranscriptomics (disease profiling) offers new opportunities to investigate which microbes are present in wild mice in an unbiased fashion by detecting any microbe (RNA virus, replicating DNA viruses, bacteria, parasite, and fungi) and identifying the actual presence of the organism, rather than a simple reflection of prior exposure history. This approach has been used in Australia on rabbits and possums [29–31].

*2.2. Physical Damage*

There is very little known about physical damage to hay and forage from the literature. There are only a few studies of house mice in hay or fodder storage systems in Australia. Two were conducted in the late 1960s [32,33], and the other was conducted by Singleton [34], who conducted a demographic study of mice in new ryegrass haystacks and mice captured in nearby fields. None of the studies measured rates of mouse damage or losses.

Newsome [32] dismantled a range of haystacks in South Australia and determined the number of mice present and signs of damage. He found that the stack's age and composition and the vertical level sampled all significantly influenced the population of mice found. More mice were found in oat stacks compared to wheat stacks; furthermore, more mice were found in bales at the top of the stack compared to bales at the bottom or middle of the stack. Newsome [32] estimated that there were up to 50 mice/ton in 9-month-old wheat/oat haystacks. He observed that mice usually ate only the embryos from the green wheat, leaving a lot of kibbling. Unfortunately, there was no overall estimate of damage measured.

Newsome and Crowcroft [33] caught about 500 mice in a few hours one night in a 4-year-old wheat haystack, the equivalent of 125 mice/ton or 5 mice/$m^3$. Mouse densities were very high, and Newsome and Crowcroft [33] found no evidence of breeding in the population, which was thought to be related to the extreme high densities and extreme lack of food.

Singleton [34] found no difference in age structure, sex, or of mean weight of mice colonising or leaving the haystack. Mice colonising the haystack appeared to be a random set of individuals from the surrounding population. However, colonisers of the haystacks survived longer and bred better than mice in the surrounding fields. Singleton [34] concluded that haystacks provided an important refuge for mice when breeding and survival in neighbouring cereal crops and margins were severely reduced. Unfortunately, damage was not measured.

When mouse populations are low, mice apparently cause relatively minor damage to fodder. During mouse plagues, mice can cause serious damage to bales and rolls (in fields or in sheds), as inferred by Newsome [32], Newsome and Crowcroft [33], and Singleton [34]. Further research is required to understand the population dynamics of mice in fodder systems, the damage they cause, and the disease risk for humans and livestock.

## 3. Mouse Control Option

There are a range of control options available to growers. Given that fodder production is often a part of the larger grain-growing and livestock enterprise, management of mice needs to be undertaken using a systems approach so that the whole farm is managed, not just the grain-growing component in isolation to hay or fodder storage.

Given the attractive nature of hay and fodder and the home-range size of mice, mouse management needs to be conducted over a large area, particularly in surrounding fields during mouse plagues, as mice will move between fodder and sheds and the surrounding fields depending on food availability. The average home range size of mice in wheat fields during the breeding season is 0.04 ha (or about 20 × 20 m). After the breeding season, home-range size increases to 0.12 ha (or about 35 × 35 m), and most mice become nomadic [35,36]. It is still important to manage habitats in and around fodder, so they are not favourable for mice. It is almost impossible to keep mice out of fodder, because they are excellent climbers, and the fodder provides excellent resources and cover/nesting sites.

It would therefore be desirable to manage surrounding fields before mice infiltrate hay and fodder storage.

The main mouse control options are through (1) rodenticides (there are limitations with using rodenticides—as discussed below), (2) physical management (barriers), and (3) other strategies. Many of these strategies are untested in an Australian context and some require significant upfront cost, so further research is required to examine the benefits of these strategies.

### 3.1. Rodenticides

Meerburg et al. [37] indicated that there are four factors that determine the uptake of a rodenticide bait: (1) whether the rodents are neophobic or neophilic, (2) the population structure of the target rodent population, (3) bait palatability, and (4) habitat structure.

In Australia, there are a range of rodenticides available for use in a broadacre perspective (against mice in crops) or in farm buildings or around houses, but there is nothing specific for use in hay and fodder storage systems. Research needs to be conducted to determine the best options available for which rodenticide could be used in and around hay bales, hay sheds, and fodder rolls. Registration of rodenticides is limited to use of zinc phosphide ($Zn_3P_2$) on crops (canola, grain crops, legumes, nuts, pasture, and safflowers; spread on the ground at a rate of 1 kg bait/ha) [38,39]. Some products (mainly the first-generation and second-generation anticoagulants) are registered for use in and around farm buildings (including grain storage; restricted to use in bait stations), domestic premises, and commercial premises. Bait stations are not currently registered for use in open fields or in and around haystacks. There are significant risks associated with secondary poisoning where anticoagulants are used [40], but this can be reduced if poisoned rodents are removed [41].

### 3.2. Physical Management

A key component of any pest management is to manage the environment to reduce pest incidence, essentially to deny them access to nesting sites and to expose them to increased predation pressure. This can be achieved through habitat management and physical management. It is often impracticable on farms to rodent-proof buildings, fodder and hay bales, and food sources, but reducing habitat complexity might disadvantage mouse populations, through increased exposure to predation (or increase predation risk) or reduced opportunities for nesting. Rodents actively avoid predators, and prefer to stay close to cover when moving between nesting and feeding sites [42]. Furthermore, mice have lower survival in areas with low grassland compared to areas with high refuge [43].

Habitat management is designed to prevent burrowing and nesting opportunities for rodents. It is almost impossible to keep mice out of fodder because fodder provides abundant food and shelter. Mice can burrow into hay bales or fodder rolls, even though they can be tightly packed, or burrow underneath them [32,33]. This means cleaning up piles of rubbish and potential mouse harbour around sheds, so the rodents cannot use it as shelter, and removing access to food. Mouse populations are known to recover quickly following reduction events through reproduction or immigration from nearby populations [44], so these actions can also help to minimise reinvasion.

There is no easy solution to managing mice once they infiltrate hay and fodder. A major problem with large-scale storage systems is the sheer scale of engineering required to make these physical barriers or structures mouse proof (especially when there are several thousand bales). Some options for physical management include:

- Robust plastic coverings: There are a range of silage tarps and hay harvest rolls on the market. Large tarps could be used to cover large stacks of hay or fodder, and individual rolls could also be covered. It is critical that these are waterproof and airtight. This would protect the bales, but importantly, an airtight covering would hopefully mean that mice do not smell the hay or fodder inside and gnaw through the plastic. Heavy-duty plastic is needed to provide a robust defence against mice because current silage wraps provide little defence against infiltration from mice.

- Burying bales: Many growers are familiar with the practice of burying silage. The advantage of this approach is that they could then be covered with large tarps or buried under soil to make it difficult for mice to dig down to the fodder. Subsequent use of the buried fodder can be logistically problematic. Fodder buried in pits needs to be removed from the pit as soon as the pit is opened. Uncovered fodder in pits is at risk of weather damage associated with significant rainfall events.

- Mouse-proof fencing around storage areas: Small metal fences could be used as a physical barrier around fodder storages. The barriers would need to be 30–50 cm high with a folded lip at the top to prevent mice from climbing over the fence. The fence would need to be set into a concrete strip or buried and back-filled with compressed road base or blue stone (out to 30–50 cm) so that mice could not dig burrows. This might be feasible in areas where hay or fodder is stored on a permanent basis. The weak link in this system is the construction of a gate, or similar, to allow access by heavy machinery, which needs to be mouse-proof.

- Build mouse-proof structures: Mouse-proof structures can be constructed with collars around the base to prevent mice climbing up onto the structure. A major limitation of this is that mice are excellent climbers, and there would be a substantial amount of engineering required to build structures over a large area.

- Shipping containers: Store hay and fodder in mouse-proof shipping containers; however, the capacity and expense of containers are limiting factors.

- Move hay/fodder to areas not affected by mice: Place hay and fodder on trucks and move it to areas where mice are not a problem to be stored, or sell it, so that other hay or fodder could be bought when needed.

*3.3. Other Strategies*

There are a range of other strategies that could be considered to manage mouse numbers in fodder storage, but these are all labour-intensive and it is unlikely they would be very successful (data on effectiveness is lacking):

- Trapping: There are many types of traps on the market, from single-capture spring traps to gas-fired piston multiple-capture traps, through to various homemade traps. Many traps are required in order to have an effect on the mouse population and appear to be ineffective when mouse populations are extremely high. In reality, these traps capture only a small proportion of the mice, especially during a mouse plague, and so are not really practical or effective in this situation. Trapping does provide an indication of a change in mouse activity, so can be a useful method for monitoring mouse populations.

- Fumigants: Might be of use in sealed or covered bales/rolls but need to be tested. As mice would die inside bales or rolls, there is still an issue of carcass disposal, because they still present a disease risk.

- Ultrasonic/magnetic devices: These might be of use only in and around hay sheds or other sheds. There are several ultrasonic and/or magnetic devices that emit electrical pulses at a range of frequencies. They might have a short-term effect on the rodents, and probably only shift the animals to another location. There are no published studies demonstrating the long-term benefit of these machines.

- Cats: Cats are not recommended because of the high risk of *Toxoplasmosis* transfer to livestock. Cats might prey on rodents, increasing predation risk (meaning rodents are less likely to move away from cover/shelter), but the disease transfer risk to livestock is simply too great.

**4. Caste Study—Survey of Mouse Damage to Fodder Storage**

*4.1. Design of Survey*

A short survey was designed to help guide a discussion with growers about mouse impacts to fodder storage. Detailed notes were taken, which formed the basis of the findings that are discussed. There were 14 surveys completed. These were conducted

with growers in areas affected by the 2021 mouse plague in NSW (Parkes, Wellington, Coonamble, Coolah, and Moree) (Figure 1) and several interviews in the Wimmera of Victoria (outside area affected by the mouse plague) (Figure 2). Currency is reported as Australian dollars (AUD).

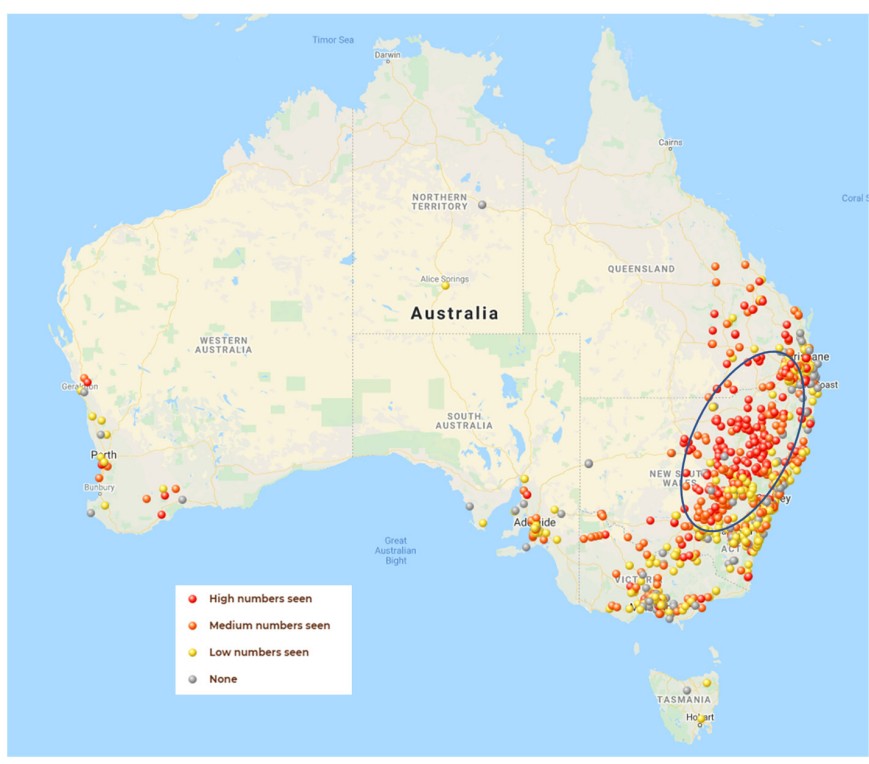

**Figure 1.** Approximate extent of the mouse plague that affected central and northern New South Wales through January to June 2021. Data sourced from *MouseAlert* web site: https://www.feralscan. org.au/mousealert/map.aspx, accessed 16 November 2021.

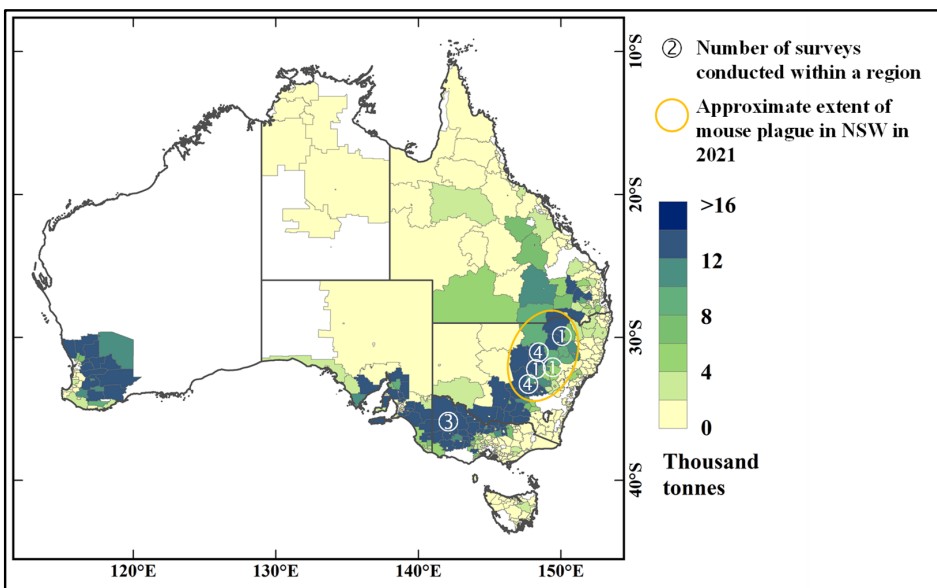

**Figure 2.** Approximate location of surveys conducted across southern and eastern Australia (*n* = 14) to assess mouse impacts on fodder storage, and approximate extent of the mouse plague that affected NSW through 2021 over a map showing fodder production. Source: Base map kindly provided by Randall Donohue (CSIRO, Land and Water) with data from Australian Bureau of Statistics [45].

All subjects gave their informed consent for inclusion before they participated in the study. The study was conducted in accordance with the Australian National Statement on Ethical Conduct in Human Research (2007 updated 2018) and has been approved and complies with the ethical research requirements of the CSIRO Social Sciences Human Research Ethics Committee (#064/21).

### 4.2. Nature of Mouse Damage

There were several common themes that emerged from the surveys, so we feel confident in our findings and that they would be broadly relevant across broad areas affected by mice.

There were different types of fodder made and stored by the respondents (oaten hay, wheaten hay, lucerne, round bales, square bales, forage, straw, etc). Most respondents kept the fodder and hay for their own purposes, as part of their mixed grain–livestock system to feed to sheep and/or cattle. There was a range of storage methods, including storage in fields, along the field edges, and in sheds. The number of bales stored ranged from hundreds to thousands of bales (with an average of around 2000 bales per grower, Table 2).

**Table 2.** Grower-reported estimates of the economic cost of managing mice in fodder storage sysTable 14. The value of bales is calculated (number of bales × value/bale), then the estimated loss by mice and the cost of control efforts (including labour) are subtracted to derive a total loss. Overall loss (%) was calculated by as a ratio of total loss/total value. Costs and losses are negative values ("−").

| Grower | State & Postcode | Value of Bales | | | | Estimated Loss by Mice | | | | | | |
|---|---|---|---|---|---|---|---|---|---|---|---|---|
| | | No. Bales | AUD/ Bale | Value (AUD) | Estimated Loss (%) | Lost Value (AUD) | Cost of Baits (AUD) | Labour | Labour Cost [1] (AUD) | Other (AUD) | Total Loss (AUD) | Overall Loss (%) |
| #1 | NSW, 2870 | 600 | 60 | **36,000** | 70% | −25,200 | −1000 | 1 day/month | −1020 | 0 | **−27,220** | **−75.6%** |
| #2 | NSW, 2870 | 370 | 50 | **18,500** | 20% | −3700 | −5000 | 1 day/week | −4080 | 0 (−300,000) [2] | **−12,780** (**−312,780**) | **−69.1%** (**−1690.7%**) |
| #3 | NSW, 2870 | 2300 | 100 | **230,000** | 30% | −69,000 | −7000 | 1 day/week | −4080 | 0 | **−80,080** | **−34.8%** |
| #4 | NSW, 2870 | 700 | 60 | **42,000** | 33% | −13,860 | −500 | 1 day/month | −1020 | 0 | **−15,380** | **−36.6%** |
| #5 | Vic, 3390 | 500 | 200 | **100,000** | 0% | 0 | −14,000 | 0 | 0 | 0 | **−14,000** | **−14.0%** |
| #6 | Vic, 3395 | 2000 | 200 | **400,000** | 10% | −40,000 | −10,000 | 0 | 0 | 0 | **−50,000** | **−12.5%** |
| #7 | Vic, 3392 | 2500 | 200 | **500,000** | 0% | 0 | −7000 | 0 | 0 | 0 | **−7000** | **−1.4%** |
| #8 | NSW, 2820 | 3500 | 250 | **875,000** | 30% | −262,500 | −25,000 | 1 day/week | −4080 | 0 | **−291,580** | **−33.3%** |
| #9 | NSW, 2829 | 0 | NA | **0** | 80% | 0 | −40,000 | 1 day/week | −4080 | −5000 | **−49,080** | **NA** |
| #10 | NSW, 2829 | 400 | 400 | **160,000** | 13% | −20,000 | −20,000 | 1 day/week | −4080 | 0 | **−44,080** | **−27.6%** |
| #11 | NSW, 2827 | 9000 | 190 | **1,710,000** | 25% | −427,500 | 0 | 1 day/week | −4080 | −30,000 | **−461,580** | **−27.0%** |
| #12 | NSW, 2828 | 2500 | 250 | **625,000** | 50% | −312,500 | −20,000 | 2 days/week | −8160 | −4000 | **−296,660** | **−47.5%** |
| #13 | NSW, 2829 | 2300 | 150 | **345,000** | 70% | −241,500 | −7000 | 2 days/week | −8160 | 0 | **−242,660** | **−70.3%** |
| #14 | NSW, 2827 | 3000 | 200 | **600,000** | 50% | −300,000 | −1000 | 1 day/month | −1020 | 0 | **−300,020** | **−50.0%** |
| **Mean** | | **2119** | **178** | **402,964** | **34%** | **−122,554** | **−11,250** | | **−3133** | | **139,723** (**−161,151**) | **−34.7%** (**−40.0%**) |

Notes: [1] We use a figure of AUD 850 per week (AUD 170/day) for labour (based on the Pastoral Award from Fair Work Australia), then multiplied by 6 to cover the 6-month period from January to June 2021 (when the mouse plague was at its peak). [2] This grower lost a tractor because of gnawing by rats and mice (AUD 300,000 value). Loss was 1690.7% if the tractor was included, or 69.1% if the tractor was excluded from the calculation. Mean total loss and mean percent loss with (in brackets) and without this calculation are also included.

Most of the respondents were using the hay/fodder for their own use, and not to sell. Many of the growers were still recovering from the drought through many regions a few years previously, and many were ensuring they were building their hay/fodder reserves to help tie themselves over the next lean period, but mice were negatively impacting this strategy. Some types of hay could be stored for 3–5 years, with some up to 10 years (if kept dry in a shed). Growers raised concerns that they may not be able to store mouse-affected hay for a long duration.

The mouse problem observed in 2021 was not usual; however, mice have not been a problem for many years (especially among the growers in NSW). The last time they were a problem was about ten years ago (likely to be after the end of the millennium drought and subsequent mouse plague through 2011/2012). Some respondents had not seen a mouse problem for approximately 30 years.

Survey respondents said that they were mostly worried about the physical damage that mice were causing to hay/fodder. Mice were regarded as the only factor (other than weather) that affected the long-term sustainability of hay storage. The sustainability of hay is affected by climatic conditions at the time of manufacture and during storage. They were concerned, but not well informed about disease transfer to livestock. Much of the damage was estimated at around 33 to 50% (mean = 34%; in several cases, up to 70%, Table 2). Cereal hay sustained significantly more damage than lucerne hay. Some of the bales were becoming loose because mice were pulling the hay out and destroying the string or netting covering the bales and they were falling apart.

More damage was observed in large sheds than bales stored in paddocks (in the open). Some of the respondents noted that mouse damage was more severe on the bales at the end of a stack of rows, with less infestation and damage on bales in the middle of rows of bales. One grower temporarily moved the end bale, then took and used the next bale to feed to his livestock and returned the more severely damaged bale to the end, so that he limited the number of bales damaged by mice. One respondent estimated that 10% of bales could not be picked up (because they would fall apart), and overall, there was 30% damage. Due to the physical damage to strings that bind the bales together and the structure of bales, transport was made impractical, and in some cases, there was total destruction of the stack because of critical damage to string. Some growers were considering burning the severely damaged bales. One grower reported burning 3000 bales of hay stored in paddocks after it was perceived to be too badly contaminated by mice to be used (a loss of AUD 120,000 based on the estimated value of hay).

Several respondents stated that cattle are selective eaters and so they use hay at a higher rate. They stated that cattle do not go near the bales when there are mice in them. They have changed their practices by feeding fodder out early to the cattle and providing more hay than they would normally feed out.

Another respondent kept fodder away from his sheds because the sheds attracted mice. He kept the fodder rolls out in his fields set out individually (his mouse infestation in the rolls was very high when we visited). He was further worried about mice accessing sheds and causing overall contamination.

Several of the growers were seeing rats in and around their hay/fodder storage but were not too concerned about them. It is not known which species they are (likely to be the brown rat *R. norvegicus*, or the black rat *R. rattus*).

*4.3. Social and Economic Impacts*

Economic impacts were accrued through a range of factors. Some of the key components included:

- Direct loss to fodder: The magnitude and cost of the impact was not well quantified. Some growers had very high levels of damage (mean loss = ~35%, up to 70% in some cases, Table 2). Bales of hay/fodder were estimated to be worth AUD 50 or AUD 60 a bale, and up to AUD 400/bale. Most growers do not plan, or were not planning, on selling their hay at the time of the survey, so it is difficult to determine the economic

loss associated with this. Additional costs will be incurred when they run out of hay/fodder and need to purchase more. Bales were falling apart because mice were infesting them, and they were not able to move them. Most hay was expected to be used within three years, but because of the mice, they must be used straight away.

- Poor efficiency of hay: Some growers were doling out twice as much hay as normal for their cattle and sheep as they are fussy eaters avoiding hay contaminated by mice, meaning they will go through their reserves twice as fast.

- Direct losses to other parts of the farming enterprise: One respondent was concerned that mice were causing all sorts of problems with his enterprise, in sheds, in the house, and in his machinery. He said he had to clean the steering wheel each time he sits in his tractor (concerned about disease) and mice had infested his seat. Another respondent recently lost an AUD 300,000 tractor to rats and mice (fire started after gnawing on electrical wires). There was also additional damage by mice to bags of pasture seed, which took extra time and labour to clean, and needed to be replaced. There were also damages to sheds and other machinery. Mice were the major problem in the grain-production component of their enterprises, and all growers were expending considerable resources in managing the mouse problem in their crops.

- Cost of mouse control: Estimates of the cost of mouse control were not well quantified. Some growers spent a few hundred dollars on managing mice through to several thousand. The main cost was for the purchase of mouse baits (principally zinc phosphide baits) (mean cost of rodenticide baits = AUD 11,250 per respondent, Table 2). Additional costs were the time required to manage the mouse problem, which ranged from one day a month to one or two days a week (average cost of ~AUD 3000 over a 6-month period, Table 2). Some growers were worried about the amount of time they were spending on managing the mouse problem when they could be doing other things around the farm that had direct economic benefit.

Using the figures provided from the respondents in this small survey, a rough calculation of estimate of losses ranged from AUD 7000 up to AUD 461,580 (mean loss = AUD 161,151 over a 6-month period of the mouse plague, Table 2). This related to a percentage overall loss of 35–40% (taking into account the value of the fodder, cost of baiting, labour, and other costs). The largest cost was associated with the loss of a tractor to fire because of rodents gnawing on wires. Losses were dependent on how many bales were held and their value (values of an individual bale ranged from AUD 50 to AUD 100, but up to AUD 400 each). It is interesting to note that in Victoria, where the mouse plague was not as serious as in NSW, the three growers incurred losses by mice mainly through the use of rodenticides (mean loss AUD 23,667 over a 6-month period, or just under 10% of the value of fodder). This was a preliminary study: the true economic impact of the mouse plague in fodder storage needs to be fully assessed through a well-designed economic impact study.

### 4.4. Concerns about Disease and Contamination

All growers were concerned about disease risks to livestock or for human health because of mice. Contamination ranged from the presence of faeces and urine to the presence of high numbers of dead and decaying mice. One grower was concerned that his animals would not eat heavily infested hay (mouse urine and faeces) because it was so rank. There was a general concern about diseases. However, the respondents did not know much about what diseases were present, what the risks were with having mice in hay/fodder storages (to humans), or how it would affect livestock (sheep or cattle). More research is required to fill this gap and to provide grower-friendly communications about the disease risks.

### 4.5. Management Practices Undertaken

The respondents knew how to manage mice in the field, but they did not really know how to manage the mouse problem in their fodder. Growers were relying on the use of rodenticides to control mice, but some were burning severely infested haystacks. Through

the 6-month period of the mouse plague, growers were spending on average around AUD 11,000 on rodenticide baits (up to AUD 25,000). Most growers were placing baits in roofs, houses, sheds, etc., and many were spreading baits around bales. Although there are no registered rodenticides for the use against mice in and around hay bales, most growers were using some type of rodenticide around their hay and fodder because of the desperate situation. Some growers used zinc phosphide (only registered for use in broadacre context in fields, not in storage situations).

Some had thought of burying hay or fodder but were concerned by the additional cost of digging a hole, burying it, then digging it back up again later (with additional labour costs). This was a similar practice to silage storage (preserved livestock food in airtight conditions), which is sometimes stored underground.

Some growers carried out some trapping, but this was limited to around sheds. Although they could catch mice each day (around 50–100), it was really a small number compared to the overall population, especially during the plague.

Some were planning to burn their hay because of the severe mouse infestation. Most growers were unsure about how to deal with mice in hay and fodder storage and could not make any suggestions about how to manage this.

## 5. General Discussion

### 5.1. Economic Damage

The economic damage caused by mice, and the cost to control mice, was poorly understood. Some fodder types were more heavily affected than others; for example, hay (without grain) was not affected as much as fodder (with grain). Most surveyed growers could not readily place a cost on the impact of mice. The survey was designed to obtain some preliminary figures to help determine some of the main economic components, but further work is required to determine this rigorously.

Our preliminary assessment provided a range of economic impacts of mice, ranging from AUD 7000 up to AUD 461,580 (mean = AUD 161,151), equating to an estimated 10% to 70% loss (mean = 35–40%). These figures were derived from a number of growers and need to be verified by scaling up to a larger number of fodder growers across the entire area affected by the mouse plague in central and northern NSW in 2021. Growers were spending one day a month and up to one to two days a week on managing mice. This is time that they could be using on more productive components of their enterprise.

More research is required to determine the economic impact of a mouse plague (severe mouse damage) versus low or moderate levels of mouse infestation in hay and fodder storage systems. An economic assessment would include the direct physical loss caused by mice, the cost of control (cost of baits, etc.), time spent managing the problem, and any other incurred costs associated with factors such as disease and contamination.

### 5.2. Disease Risk

Growers were somewhat concerned about disease risk but were not aware of the specifics of rodent-borne diseases (or parasites) or their impacts on livestock or humans. A major gap in our knowledge is the disease and parasite burden of mice in hay and fodder systems, and how they interact with livestock. A full assessment of diseases needs to be determined from mouse populations, hay and fodder samples, and livestock. Surveys of mice from different population densities (low, moderate, and high) to understand the role and burden of diseases and parasites is required.

There is little information on which pathogens are present in Australian wild mice at the population level and how pathogen composition varies with changes in mouse population abundance. With continued improvements in DNA sequencing technologies, metatranscriptomics analysis (disease profiling) offers new opportunities to investigate which microbes are present in wild mice in Australia. Metatranscriptomics involves sequencing the total RNA present in a sample, which can identify RNA viruses, as well as replicating DNA viruses, bacteria, parasites, and fungi. This approach is unbiased in

that any microbe at sufficiently high abundance will be detected, and it identifies the actual presence of the organism, rather than simply reflecting prior exposure history.

*5.3. Managing Mice in Fodder*

There is a lot of knowledge available about how to control mice in grain-based farming systems (in a broadacre context), but very little is known about what to do and what is effective in hay and fodder storage systems. Some growers were changing their practices in response to mouse infestation (particularly in the areas affected by the mouse plague in NSW) through feeding out extra hay or fodder (at twice the rate) to account for livestock rejection of contaminated sections of hay or fodder.

It is important to determine what control practices are relevant for forage storage, test them, and quantify their effectiveness. As outlined earlier, there are a range of control options available (physical barriers, rodenticides, other methods), but these need to be experimentally tested and assessed for economic benefits. There is not much in the general scientific literature about control of mice in hay or forage systems, apart from a few papers published in the 1970s and 1980s. As stated earlier, even the definitive "Rodent Pests and their Control" [13] does not contain any information about impacts of mice in hay or fodder systems, or what control options are available.

Research needs to be directed toward establishing what control options are available and how effective they are. Some of the main questions about control options are as follows:

1.  What mouse control strategies should be conducted in low mouse years (normal phase)? What routine practices should be recommended each year, especially when numbers are not high? Is there any benefit of controlling mice in "low" years so that their impact is not as great in "high" years?
2.  What mouse control strategies should be conducted when mouse populations are increasing or in "high" mouse years (consideration of preventative measures versus emergency response)?
3.  How effective are a range of different management practices? What are the benefits of covering bales or fodder rolls, burying, using other physical barriers, baiting, or burning fodder or hay bales?

## 6. Conclusions

The impacts of mice to fodder in Australia are not well understood. We conducted a pilot survey of growers ($n$ = 14) to determine some of the economic impacts, to understand disease risk to livestock and humans and explore management practices being used at a time when a mouse plague occurred in New South Wales, Australia. Research is required to (1) determine the economic impact of mice in hay and fodder systems (especially during mouse plagues); (2) undertake a disease study of mice, as well as what diseases are present in the hay and fodder itself, to help inform appropriate management strategies to control mice, and potentially better manage hay and fodder to eliminate disease transfer to livestock and humans; and (3) identify effective mouse control options for hay and fodder systems, including identifying rodenticides, physical barriers, and other strategies, and to determine the economic benefits.

**Author Contributions:** P.R.B. and S.H. conceived and carried out the work. P.R.B. wrote the paper with input from S.H. All authors have read and agreed to the published version of the manuscript.

**Funding:** This project was supported by the National Recovery and Resiliency Agency, The Department of Prime Minister and Cabinet (Project No. DRD000113), as well as CSIRO Health & Biosecurity.

**Institutional Review Board Statement:** This research has been approved and complies with the ethical research requirements of the CSIRO Social Sciences Human Research Ethics Committee (#064/21).

**Informed Consent Statement:** Informed consent was obtained from all subjects involved in the study.

**Data Availability Statement:** The data that support this study will be shared upon reasonable request to the corresponding author.

**Acknowledgments:** We sincerely thank Thomas Boulton and Bas Wilson (DPM&C and NRRA) for their support and encouragement with this project. We also acknowledge the support from Leigh Nelson (GRDC) who has assisted with this work. We thank Randall Donohue (CSIRO, Land and Water) for providing the base map used in Figure 2. We are indebted to the growers for their participation in the survey. We also thank Krista Verlis, Michael Davies, and Wendy Ruscoe for constructive comments on a draft version of this paper.

**Conflicts of Interest:** The authors have no conflict of interest to declare.

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
