# Peer review of "Impacts of House Mice on Sustainable Fodder Storage in Australia"

_agronomy, doi:10.3390/agronomy12020254_

Round 1

Reviewer 1 Report

This manuscript brings a literature review on quantification of damages on fodder storage caused by house mouse in Australia. Further, authors performed a pilot survey to determine the range of damages and management strategies in fodder growers. In Australia region, house mice are pests both in the field and synanthrophic area and exhibit there periodic outbreaks with enormous numbers. This phenomenon is unique, as most of world regions consider house mice as pests in urban or agriculture environment, not in crops. The topic of rodent damages on fodder is understudied not only in Australia, but also in the rest of world. Moreover, the topic of rodent damages themselves belongs to the understudied scientific topics in general…

I have only the following minor comments to the text:

Line 96: Mites are mentioned twice in the parentheses.

Table 2: It is very hard to follow this table (because of typesetting). If it is a final form, it should be changed to improve the readability. If not, there are still some points need to revise:

  • Number index 1,2 are not explained
  • Using of “-“ is little bit confusing in this form of table as sometimes it means negative values and sometimes the missing values. Maybe/probably, only the narrow columns and corresponding dividing the text into two rows are the reason of this confusion. I am not able to review the table in this form.

It is repeatedly mentioned in the text that livestock avoided the hay contaminated by mice faeces/urine (line 413-415, 451). This phenomenon was described earlier by Daniels and Hutchings (2001), it would be useful to add this reference to “2. Potential impacts of rodents on damages” – line 87.

Daniels, M.J. and Hutchings, M.R. (2001) The response of cattle and sheep to feed contaminated with rodent faeces. The Veterinary Journal 162(3), 211–218.

Line 532-541: Remove double numbering of paragraphs

Author Response

The authors thank the reviewer for the comments provided on the manuscript. We have made all the relevant changes to the manuscript as request.

We respond to each comment below:

Line 96: Mites are mentioned twice in the parentheses. Response: the second "mites" was deleted.

Table 2: It is very hard to follow this table (because of typesetting). If it is a final form, it should be changed to improve the readability. If not, there are still some points need to revise:

  • Number index 1,2 are not explained
  • Using of “-“ is little bit confusing in this form of table as sometimes it means negative values and sometimes the missing values. Maybe/probably, only the narrow columns and corresponding dividing the text into two rows are the reason of this confusion. I am not able to review the table in this form.

Response: We don't know what happened to the the layout - it looked good when we submitted the manuscript. We have revised the table (made it landscape) so there are no text carried over to multiple lines. Apologies the footnotes for 1,2 were omitted when transfering text into the journal template (they have been reinstated). Missing values are now reported as $0 if there was no cost and "NA" if not applicable.

It is repeatedly mentioned in the text that livestock avoided the hay contaminated by mice faeces/urine (line 413-415, 451). This phenomenon was described earlier by Daniels and Hutchings (2001), it would be useful to add this reference to “2. Potential impacts of rodents on damages” – line 87. Response: Many thanks for this reference - it is highly relevant. We have included it as suggested.

Daniels, M.J. and Hutchings, M.R. (2001) The response of cattle and sheep to feed contaminated with rodent faeces. The Veterinary Journal 162(3), 211–218. 

Line 532-541: Remove double numbering of paragraphs. Response: This was an error when the text was converted into the journal template, which affected dot points and numbered lists. We have corrected bullet numbering throughout.

Reviewer 2 Report

This research collects new information to better understand the losses and damage caused by mice to fodder systems in Australia. This contribution is significant as there are gaps in the existing knowledge on the impacts of mice to fodder storage, being useful to inform appropriate management strategies for effective control, as well as minimize damage, losses and disease risks for humans and livestock.

This is a novel, well-structured and well-written work. The design of survey is appropriate.

Author Response

We thank the reviewer for thier kind comments. There is no need to change the manuscript based on these comments.